# Performance-Status Deterioration during Sequential Chemo-Radiotherapy as a Predictive Factor in Locally Advanced Non-Small Cell Lung Cancer

**Marta Olszyna-Serementa** [1], **Magdalena Zaborowska-Szmit** [1], **Sebastian Szmit** [2,3,*], **Piotr Jaśkiewicz** [1], **Katarzyna Zajda** [1], **Maciej Krzakowski** [1] **and Dariusz M. Kowalski** [1]

1 Department of Lung Cancer and Thoracic Tumors, Maria Sklodowska-Curie National Research Institute of Oncology, 02-781 Warsaw, Poland
2 Department of Cardio-Oncology, Centre of Postgraduate Medical Education, Institute of Hematology and Transfusion Medicine, 02-776 Warsaw, Poland
3 Clinic of Oncological Diagnostics and Cardio-Oncology, Maria Sklodowska-Curie National Research Institute of Oncology, 02-781 Warsaw, Poland
* Correspondence: s.szmit@gmail.com

**Abstract:** The role of sequential chemoradiotherapy in non-small cell lung cancer (NSCLC) patients who are not eligible for concurrent therapy has not been clearly defined. The aim of this study was to determine the usefulness of Karnofsky performance status (KPS) monitoring and to define the factors determining clinical deterioration during sequential chemoradiotherapy in patients treated from July 2009 to October 2014. The study included 196 patients. The clinical stage was defined as III A in 94 patients (48%) and III B in 102 patients (52%). Reduced KPS was found in 129 patients (65.8%). Baseline KPS had no significant prognostic significance. Deterioration of KPS during chemoradiotherapy was observed in 53 patients (27%) and had a negative predictive value for both worse-progression free survival (HR = 1.44; 95% CI: 1.03–1.99; $p = 0.03$) and overall survival (HR = 1.42; 95% CI: 1.02–1, 99; $p = 0.04$). The deterioration of KPS correlated with the disease control rate 6 weeks after the end of chemoradiotherapy ($p = 0.0085$). The risk of KPS worsening increased with each subsequent day between the end of chemotherapy and the start of radiotherapy (OR = 1.03; 95%CI: 1.01–1.05; $p = 0.001$), but decreased with each year of older age of patients (OR = 0.94, 95% CI: 0.9–0.98, $p = 0.009$). The time between the end of chemotherapy and the start of radiotherapy determined the prognosis of NSCLC after chemoradiotherapy. It should be adjusted to the age of patients.

**Keywords:** performance status; chemoradiotherapy; non-small cell lung cancer; prognosis

## 1. Introduction

Assessment of performance status is an important part of the prognostic stratification of any cancer patient. One of the scales used is the Karnofsky Performance Status (KPS) scale. Worse KPS is determined by many factors, such as heart dysfunction, limitation in ventilation, decreased lung capacity, significant weight loss, anemia, morbid obesity or cachexia, neurological impairment, and older age.

Performance status is a very important factor taken into account by oncologists when choosing anticancer treatment. An example of such a decision is the choice between concurrent and sequential chemoradiotherapy in locally advanced unresectable lung cancer. However, there are no studies on how the deterioration of performance status during quite long-term treatment with chemoradiotherapy or other anticancer treatment regimens affects the long-term prognosis.

The aim of this study was to determine the usefulness of monitoring performance functional status and the factors determining its deterioration in patients with unresectable non-small cell lung cancer (NSCLC) undergoing radical sequential chemoradiotherapy.

## 2. Methods

The retrospective, uni-institutional, observational study as a PhD project was planned for identification of all possible prognostic and predictive factors in locally advanced NSCLC patients treated by radical chemoradiotherapy. The presented study included patients who underwent sequential chemoradiotherapy in the years 2009–2014 in the largest Polish oncology center. Data of long-term clinical observation were used in the analysis of predictors for progression-free and overall survival (PFS, OS).

Inclusion criteria for sequential rather than concurrent chemoradiotherapy were defined according to local guidelines of National Research Institute of Oncology in Warsaw. Each patient had to have at least one of the following characteristics: (A) older age, which was confirmed as a favorable choice for this group of patients [1], (B) significant comorbidities, and/or (C) reduced exercise capacity defined below 100 by the Karnofsky Performance Scale (KPS).

Assessment of KPS was mandatory, at least at the start of treatment before chemotherapy and at the end of radiotherapy. KPS deterioration was understood as a decrease of at least 10 points between the first and last assessment.

Radiosensitizing chemotherapy had to be used before radiotherapy; there was a possible choice between (1) cisplatine-based regimen (PN: cisplatin + vinorelbine or PG: cisplatin + gemcitabine or PE cisplatin + etoposide) and (2) carboplatin-based regimen (carboplatin + vinorelbine or carboplatin + paclitaxel). The administered doses of radiation therapy were in the range of 5880 cGy to 6600 cGy.

The toxicity of chemotherapy and radiotherapy was recognized in accordance with the Common Terminology Criteria for Adverse Events (CTCAE).

The primary observation point was overall survival (OS). This was the time from the start of chemotherapy to the moment of death from any cause. The secondary endpoint was progression-free survival (PFS), which was measured from the date of the initiation of chemotherapy to the date of disease progression or death (if no progression was previously observed). The Kaplan–Meier estimator and Cox proportional hazard analysis were used for the evaluation of relationships between baseline KPS and the deterioration of KPS with OS and PFS. The odds ratio was used to assess risk factors for the occurrence of KPS deterioration during chemoradiotherapy.

## 3. Results

The final analysis of the study included a cohort of 196 patients with locally advanced unresectable NSCLC. The following patients met the inclusion criteria: 52 patients (26.53%) were over 65 years of age; 140 patients (71.43%) had a high Charlson Comorbidity Index (CCI > 3); and 129 patients (65.82%) had decreased baseline Karnofsky performance status (KPS < 100) [2]. The demographic characteristics of the whole cohort are presented in the Table 1.

**Table 1.** Characteristics of patients included to the study.

| Parameters | |
|---|---|
| Sex | Female—69 (35.2%)<br>Male—127 (64.8%) |
| Age (years) | $60.52 \pm 7.1$ |
| BMI (kg/m$^2$) | 26.5 (IQ: 23.2–30.4) |
| Lack of weight loss | 107 (54.6%) |
| Weight loss <10%<br>Weight loss ≥10% | 65 (33.2%)<br>24 (12.2%) |

**Table 1.** *Cont.*

| Parameters | |
|---|---|
| Pathology | |
| Squamous-cell carcinoma | 91 (46.4%) |
| Adenocarcinoma | 47 (24.0%) |
| Other types | 58 (29.6%) |
| Advancement of cancer disease | Clinical Stage III A—94 (48%) |
| | Clinical Stage III B—102 (52%) |
| | T4—64 (32.7%) |
| | N3—20 (10.2%) |
| Declared smoking status | |
| never smokers | 16 (8.16%) |
| <20 pack years | 41 (20.92%) |
| 20–50 pack years | 94 (47.96%) |
| ≥50 pack years | 45 (22.96%) |
| Charlson Comorbidity Index (CCI) | |
| ≤3 | 56 (28.6%) |
| 4 | 55 (28.1%) |
| 5 | 44 (22.4%) |
| ≥6 | 41 (20.9%) |
| Baseline Hemoglobin (HGB, g/dL) | 13.7 (IQ: 12.7–14.5) |
| Baseline renal function expressed according to the Cockcroft-Gault equation (mL/min.) | 83 (IQ: 69–104) |
| No complications | 36 (18.4%) |
| Serious complications (grade 3 or 4) | 67 (34.2%) |

Among the 196 patients qualified for analysis to determine the prognosis of patients undergoing sequential chemoradiotherapy, only about 1/3 of patients (67 patients, 34.2%) obtained the highest score, i.e., 100 according to the Karnofsky Performance Scale (KPS) before treatment. The remaining patients had lower KPS: 90 in 101 patients (51.5%) and 80 in 28 patients (14.3%).

Baseline good performance status, defined as 100 by KPS, was not significant for predicting progression-free survival (HR = 1.02; 95% CI: 0.75–1.40; $p$ = 0.88) or overall survival (HR = 0.96 95% CI: 0.69–1.32, $p$ = 0.79).

At the end of anticancer treatment, i.e., after radiotherapy, deterioration of KPS was observed in 53 patients (27.0%), including:

1. worsening by 10 in 41 patients (20.9%),
2. worsening by 20 in 10 patients (5.1%),
3. worsening by 30 in 2 patients (1.0%).

Patients with observed deterioration of KPS during sequential chemoradiation had, in long-term follow-up:

- 42% significantly higher risk of mortality (HR = 1.42; 95% CI: 1.02–1.99; $p$ = 0.04) (Figure 1).
- 44% significantly higher risk of cancer progression (HR = 1.44; 95% CI: 1.03–1.99; $p$ = 0.03) (Figure 2).

The advancement of NSCLC expressed by the clinical stage, T4, or N3 feature was not significant for the deterioration of performance status (Table 2). Patients over 65 years of age had a significantly reduced risk of KPS deterioration (OR = 0.4; 95% CI: 0.17–0.92; $p$ = 0.03) during sequential chemoradiotherapy. The chance of performance-status deterioration was significantly lower—by 6% (OR = 0.94; 95% CI: 0.9–0.98; $p$ = 0.009) with each successive year of patients' age. Moreover, the better the patient's baseline performance status, the greater the observed risk of its deterioration during chemoradiotherap—by 25% for every 10 on the Karnofsky scale (OR = 1.25; 95% CI: 1.17–1.35; $p$ < 0.00001). If the patient had a

KPS of 100 at baseline, he or she had a several-fold increased risk of performance-status deterioration (OR = 13.22; 95% CI: 6.2–28.2; $p$ < 0.00001) during treatment with sequential chemoradiotherapy.

The worsening of KPS was not affected by comorbidities or even by weight loss as a direct complication of cancer (Table 3).

An extremely important finding was that the risk of performance-status deterioration increased with each day of the interval between the end of chemotherapy and the start of radiotherapy (OR = 1.03; 95% CI: 1.01–1.05; $p$ = 0.001). If this time was shorter than 19 days, then the risk of performance-status deterioration was four times lower (OR = 0.25; 95% CI: 0.09–0.68; $p$ = 0.006). If this time interval was longer than 42 days (6 weeks), the risk of performance-status deterioration was more than doubled (OR = 2.21; 95% CI: 1.09–4.5; $p$ = 0.028) (Table 4). KPS worsening did not depend on the duration of chemotherapy, the number of cycles of chemotherapy, the type of chemotherapy regimen, or the duration of radiotherapy (Table 4).

The occurrence of individual complications did not significantly correlate with the deterioration of performance status (Table 5). However, patients who did not experience complications had a significantly lower chance of performance-status deterioration (OR = 0.28; 95% CI: 0.09–0.85; $p$ = 0.02).

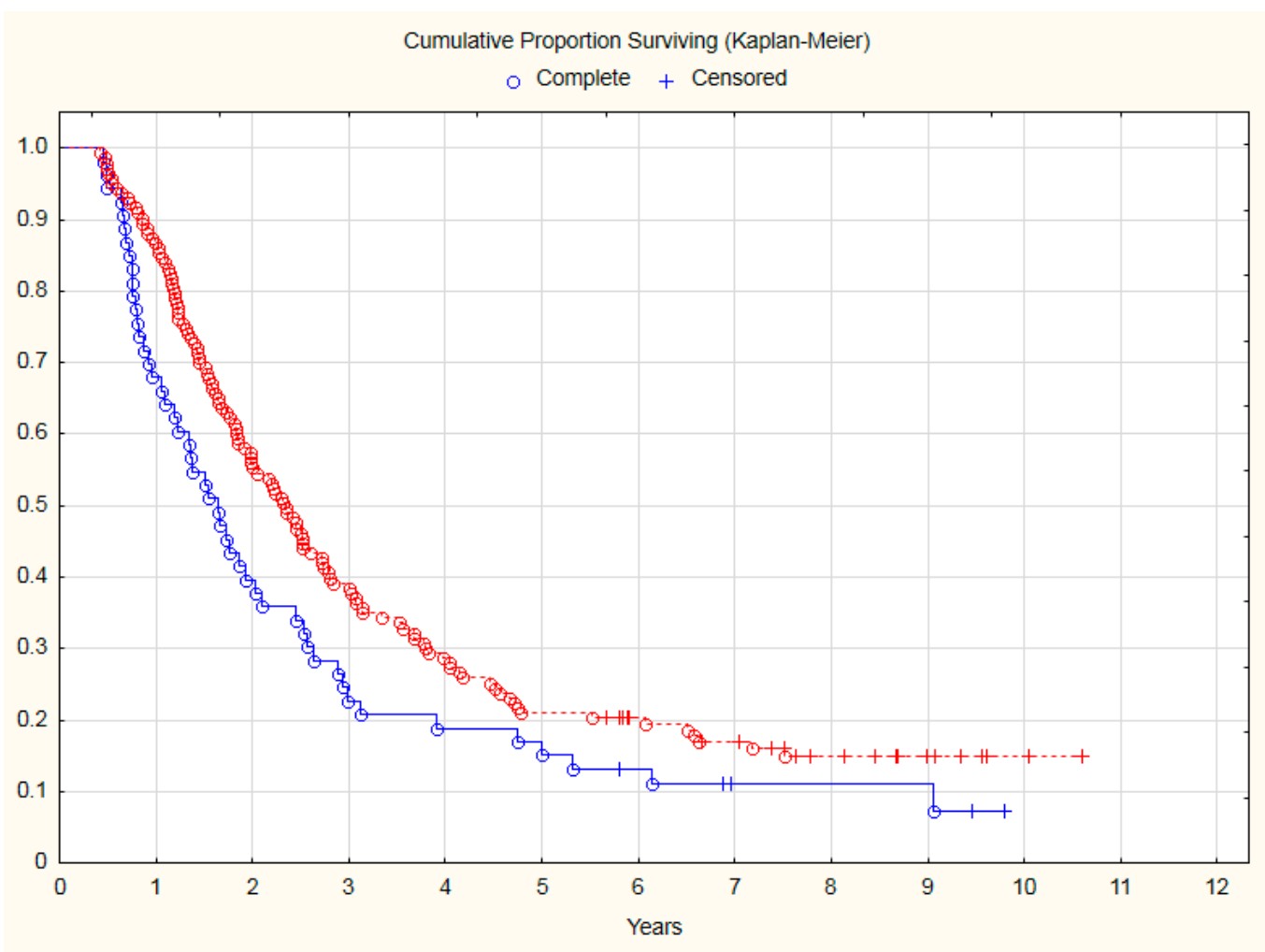

**Figure 1.** The comparison of OS between NSCLC patients with KPS deterioration during chemoradiotherapy (blue line) and patients with stable performance status (red line). Median of OS: 18.77 months vs. 28.23 months. Risk of mortality: HR = 1.42; 95% CI: 1.02–1.99; $p$ = 0.04.

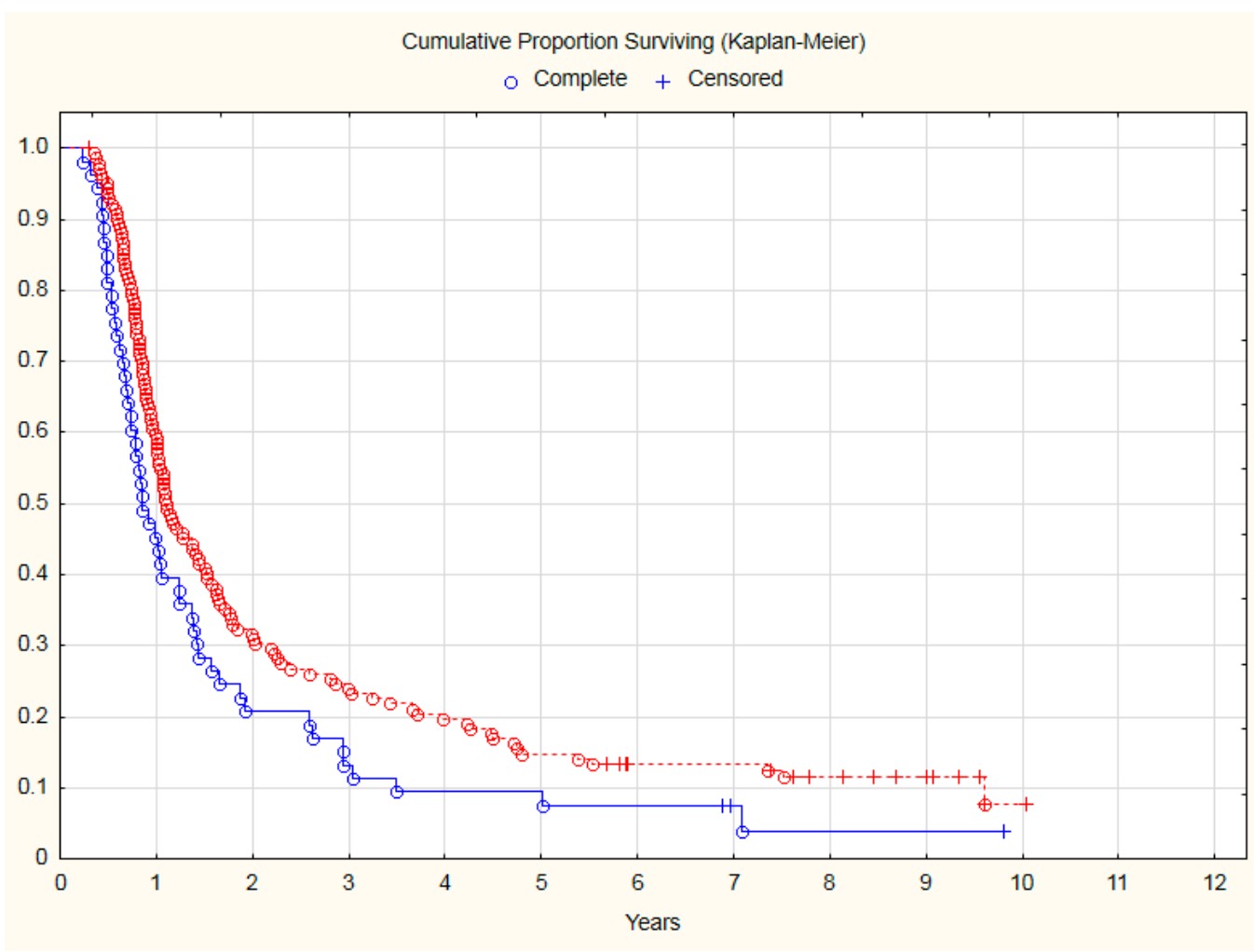

**Figure 2.** The comparison of PFS between NSCLC patients with KPS deterioration during chemoradio-therapy (blue line) and patients with stable performance status (red line). Median of PFS: 10.27 months vs. 13.33 months. Risk of NSCLC progression: HR = 1.44; 95% CI: 1.03–1.99; *p* = 0.03.

**Table 2.** Demographic and cancer-related factors affecting the deterioration of KPS during sequential chemotherapy and radiotherapy.

| | Prognostic Factor | OR | 95% CI | *p* |
|---|---|---|---|---|
| Basic demographics | Female | 0.93 | 0.48–1.81 | 0.82 |
| | Age over 65 years | 0.4 | 0.17–0.92 | 0.03 |
| | Age (each year of seniority) | 0.94 | 0.9–0.98 | 0.009 |
| | Obesity (BMI ≥ 30 kg/m$^2$) | 0.78 | 0.37–1.65 | 0.51 |
| History of smoking | Never | 2.02 | 0.72–5.66 | 0.18 |
| | ≥20 pack-years | 0.82 | 0.41–1.63 | 0.57 |
| | ≥50 pack-years | 0.98 | 0.46–2.07 | 0.95 |
| Performance status | Baseline KPS (each 10) | 1.25 | 1.17–1.35 | <0.00001 |
| | Baseline KPS = 100 | 13.22 | 6.20–28.2 | <0.00001 |
| Histopathological diagnosis | Squamous cell carcinoma | 1.42 | 0.75–2.69 | 0.27 |
| | Adenoracinoma | 1.04 | 0.5–2.19 | 0.91 |
| | Others | 0.62 | 0.3–1.29 | 0.2 |

**Table 2.** *Cont.*

| | Prognostic Factor | OR | 95% CI | *p* |
|---|---|---|---|---|
| | State IIIA vs. IIIB | 1.31 | 0.69–2.47 | 0.41 |
| Advancement of cancer disease | T4 | 0.96 | 0.49–1.9 | 0.92 |
| | N3 | 0.65 | 0.2–2.05 | 0.46 |

OR—odds ratio, which determines the chance of KPS deterioration; a value above 1 determines how much greater (a value below 1 determines how much lower) the chance is of KPS deterioration occurring in patients with a particular prognostic factor.

**Table 3.** Direct complications related to NSCLC and comorbidities as potential predictors of worsening of KPS during sequential chemotherapy and radiotherapy.

| | Prognostic Factors | OR | 95% CI | *p* |
|---|---|---|---|---|
| History of weight loss | No weight loss | 1.12 | 0.59–2.12 | 0.73 |
| | weight loss $\geq$ 10% | 0.35 | 0.1–1.23 | 0.1 |
| Hemoglobin (HGB) | HGB < 12.7 g/dL | 0.78 | 0.36–1.68 | 0.52 |
| | HGB > 14.5 g/dL | 1.0 | 0.46–2.19 | 0.99 |
| History of acute CV events | Arterial thromboembolic events (e.g., myocardial infarction) | 0.57 | 0.22–1.49 | 0.25 |
| | Venous thromboembolic diseases (e.g., pulmonary embolism) | 1.08 | 0.2–5.81 | 0.93 |
| History of internal diseases | Diabetes | 1.28 | 0.46–3.57 | 0.64 |
| | Chronic obstructive pulmonary disease | 0.55 | 0.23–1.28 | 0.16 |
| | Thyroid diseases | 0.65 | 0.2–2.05 | 0.48 |

OR—odds ratio, which determines the chance of KPS deterioration; a value above 1 determines how much greater (a value below 1 determines how much lower) the chance of KPS deterioration occurring in patients with a particular prognostic factor.

**Table 4.** Characteristics of sequential chemotherapy and radiotherapy in relation to KPS worsening. The values of the lower and upper quartiles were used as prognostic factors.

| | Prognostic Factor | OR | 95% CI | *p* |
|---|---|---|---|---|
| Duration of chemotherapy | <28 days | 0.94 | 0.37–2.37 | 0.89 |
| | >53 days | 1.32 | 0.64–2.7 | 0.45 |
| Number of cycles of chemotherapy | <2 | 0.37 | 0.04–3.15 | 0.36 |
| | >3 | 2.57 | 0.88–7.52 | 0.08 |
| Chemotherapy regimen | Cisplatin + vinorelbine | 1.24 | 0.54–2.85 | 0.6 |
| | Chemotherapy without cisplatin | 0.42 | 0.12–1.49 | 0.18 |
| The time between the end of chemotherapy and the start of radiotherapy <19 days >42 days | | 0.25 2.21 | 0.09–0.68 1.09–4.5 | 0.006 0.028 |
| Each consecutive day between chemotherapy and radiotherapy | | 1.03 | 1.01–1.05 | 0.001 |
| Duration of radiotherapy | <28 days | 1.66 | 0.61–4.5 | 0.32 |
| | >45 days | 1.1 | 0.48–2.49 | 0.82 |

OR—odds ratio, which determines the chance of KPS deterioration; a value above 1 determines how much greater (a value below 1 determines how much lower) the chance of KPS deterioration occurring in patients with a particular prognostic factor.

**Table 5.** Complications and worsening of KPS during sequential chemotherapy and radiotherapy.

| | Prognostic Factors | OR | 95% CI | *p* |
|---|---|---|---|---|
| No complications | | 0.28 | 0.09–0.85 | 0.02 |
| Complications of any degree | Pneumotoxicity | 1.75 | 0.71–4.29 | 0.22 |
| | Cardiotoxicity | 0.98 | 0.29–3.26 | 0.97 |
| | Nephrotoxicity | 1.01 | 0.37–2.75 | 0.98 |
| | Neutropenia | 0.76 | 0.38–1.52 | 0.43 |
| | Anemia | 0.8 | 0.21–3.04 | 0.74 |
| | Esophagitis | 1.63 | 0.8–3.33 | 0.18 |
| | Neuropathy | 1.66 | 0.38–7.25 | 0.5 |
| Serious complications | Any grade 3 or 4 complication | 1.38 | 0.72–2.67 | 0.33 |
| | Grade 3 or 4 neutropenia | 0.98 | 0.46–2.07 | 0.95 |
| | Febrile neutropenia | 0.9 | 0.17–4.63 | 0.89 |
| | Acute renal injury | 2.25 | 0.58–8.81 | 0.24 |
| | Grade 3 or 4 pulmonary complications | 1.83 | 0.29–11.4 | 0.51 |
| | Grade 3 or 4 cardiac complications | 4.23 | 0.68–26.36 | 0.12 |

OR—odds ratio, which determines the chance of KPS deterioration; a value above 1 determines how much greater (a value below 1 determines how much lower) the chance of KPS deterioration occurring in patients with a particular prognostic factor.

First, the objective response rate (ORR) was evaluated by computed tomography 6 weeks after the end of radiotherapy. The complete response (CR) was found in 15 patients (7.7%) and partial response (PR) in next 123 patients (62.8%). There was a statistically borderline correlation between ORR and deterioration of KPS ($p = 0.06$). Stable disease (SD) was confirmed in next 44 patients (22.4%) and progressive disease (PD) in 14 patients (7.1%). Finally, the disease control rate (CR+PR+SD) significantly correlated with the deterioration of KPS (Table 6).

**Table 6.** The correlation between deterioration of KPS and disease control rate 6 weeks after the end of chemoradiotherapy.

| | Complete Response or Partial Response or Stable Disease | Progressive Disease | |
|---|---|---|---|
| KPS worsening | 45 | 8 | Chi-square = 6.92 |
| Stable KPS | 137 | 6 | $p = 0.0085$ |

## 4. Discussion

Low exercise activity is associated with an increased probability of cardiovascular disease (CVD) or cancer development [3]. Low exercise tolerance is also associated with increased all-cause, cancer-related, and CVD-related mortality [4,5]. Physical activity may have beneficial effects in cardiology and oncology [6,7]. Even in advanced cancer, physical activity can not only improve the quality of life but also prolong survival [8].

In lung cancer, the decision to use radical treatment is conditional upon the state of cardiopulmonary efficiency [9]. In addition, through personalized physical exercises, the risk of perioperative complications can be reduced [10].

Cancer disease can lead to a deterioration in performance status; in the case of lung cancer, there is a clear cause-and-effect relationship, because reduced performance status correlates with changes in heart function and unfavorable prognosis [11]. Cancer treatment may also contribute to performance-status/exercise-tolerance deterioration by inducing a

variety of quality-of-life toxicities. An example is the cardiotoxicity seen in the treatment of breast cancer [12].

The effectiveness of treatment of locally advanced lung cancer is still unsatisfactory, and the use of concurrent chemoradiotherapy is often impossible due to the poor performance status and the expected high risk of complications. The randomized phase III trial RTOG 9410 showed that concurrent chemoradiotherapy, in comparison with sequential chemoradiotherapy, leads to higher rates of acute grade 3–5 nonhematologic toxic effects, but late complications were similar [13]. Therefore, there is a constant need for research into the role of sequential chemoradiotherapy, both in terms of effectiveness and cost [14].

Many studies have previously indicated that concurrent chemoradiotherapy is associated with greater toxicity, compared with sequential chemoradiotherapy; therefore, it should be reserved for younger patients, patients with minimal weight loss, and patients in good performance status [15]. Based on our own experience, we recommend a very thorough assessment of prognostic factors in each patient to select the optimal treatment strategy—concurrent or sequential. During selection, the following factors were defined as prognostic and important for decision-making: performance status ($p < 0.001$) and weight loss ($p < 0.001$), and (with less statistical significance) age ($p < 0.05$), tumor size ($p < 0.05$), and lymph node involvement ($p < 0.05$).

One of the earlier studies showed that, apart from the advancement of the cancer disease, anemia, index of comorbidities, and the assessment of the quality of life based on the patient Lung Cancer Symptoms Scale (LCSS) may have significant independent prognostic value [16]. The scale included the assessment of fatigue, cough, shortness of breath, physical activity, haemoptysis, pain, appetite disorders, and feelings of stress. According to the authors, an indirect explanation of clinical usefulness may be the correlation of this scale with the patient's performance status.

Historically, in inoperable locally advanced lung cancer, five groups of patients with significantly different prognoses of survival have been identified [17]. The model was built on the basis of the results of nine clinical trials; the key role was assigned to performance status according to KPS (above or below 90). Then, age and the presence of pleural effusion were significant. Chemotherapy improved prognosis only in patients with good KPS. Accordingly, many authors believe that only patients with good performance status gain the greatest benefit from concurrent chemoradiotherapy [18].There are opinions that the daily life activity of a patient with locally advanced lung cancer may predict complications during radiotherapy (risk of hospitalization due to complications and prolongation of treatment) [19].

The data presented in this article provide a new perspective on the importance of performance status in oncology. For the prediction of PFS and OS after sequential chemoradiotherapy, it was not significant whether a patient had a baseline KPS of 100 or less. The key factor for the prognosis was whether the performance status deteriorated during anticancer treatment. Reported new symptoms of exercise intolerance and dyspnoea were significantly associated with unbeneficial PFS and OS. As in many other studies, the effects of the toxic effect of chemoradiotherapy on the state of comorbidities, especially in the cardiovascular and respiratory systems, could be expected. Meanwhile, the worsening of KPS was not associated with comorbidities in our study. The risk of KPS worsening was also not greater in older patients. Younger patients experienced a worsening of KPS more often, and patients with good baseline performance more often experienced a decrease in quality of life during chemoradiotherapy. The explanation for this was the observation that the risk of KPS worsening correlated with each successive day between the end of chemotherapy and the start of radiotherapy. So far, it has been proven in the literature that the duration of radiotherapy is of prognostic importance [20]. The presented results show that the time between chemotherapy and radiotherapy may be more important. Every day seems to be essential for prognosis and this should be kept in mind during logistic organizing sequential chemoradiotherapy.

The presented data show that sequential chemoradiotherapy is not the optimal treatment for younger patients (age below 65 years) and patients with good baseline performance (KPS equal to 100). These patients have a significantly higher risk of worsening KPS, which means that already during chemoradiotherapy the quality of life deteriorated. The reason was ineffective cancer therapy in these patents, the final consequence was unfavorable PFS and OS.

In advanced lung cancer, deterioration in performance status does not necessarily correlate with tumor progression in the short-term follow-up [21]. However, the cardiotoxicity of both radiotherapy and anticancer pharmacotherapy may result in a deterioration of exercise tolerance [22,23]. A decrease in exercise tolerance may be a direct consequence of the administered dose of radiotherapy involving the heart [24]. Some studies suggested physical training to prevent the toxicity of concurrent or sequential chemoradiotherapy [25]. Recommendations for cardiac rehabilitation can also be found in cardio-oncology guidelines [26]. The long-term effects of chemotherapy and radiotherapy can manifest in fatigue and shortness of breath [27].

In our observation, there was no relationship between the occurrence of complications and the worsening of KPS in the short-term follow-up. The deterioration of KPS was surely the result of insufficiently intensive anticancer treatment, especially in younger patients with baseline-better KPS. The fact that uncomplicated patients had a lower chance of worsening KPS reflects that the intensity of treatment was appropriately adjusted in this subgroup of patients.

Our results should become the basis for the creation of a personalized qualification algorithm for concurrent or sequential chemoradiotherapy, taking into account the performance status determined by the advancement of the cancer disease, patients' ages, and status of comorbidities. There are examples in other cancers, where attempts are made to personalize the intensity of treatment by selecting doses of cytostatics and minimizing the intervals between chemotherapy courses, which is of predictive and prognostic importance, asin lymphomas [28]. Similar personalization should apply to sequential chemoradiotherapy, i.e., for patients receiving sequential therapy instead of concurrent therapy, determining what chemotherapy regimen and what interval between chemotherapy and radiotherapy are appropriate.

Some key points are important to highlight. Our observation is a hypothetical study. The hypothesis assumes that the deterioration of KPS is due to the lower effectiveness of cancer treatment. This is confirmed by the fact that the deterioration of KPS correlates with each day of treatment extension, and precisely with each day between the end of chemotherapy and the start of radiotherapy. Our data clearly show that the younger the patient and the better the initial performance status, the greater the chance of worsening KPS because sequential chemoradiotherapy with a long time between chemotherapy and radiotherapy is not a sufficiently intensive cancer treatment for such patients.

The oncologists decided about a delay in starting radiotherapy. Among the most important reasons were that the patients did not tolerate the chemotherapy or already had a significant reduction in KPS before radiation. Such a situation can explain the fact that each day of delay in radiation portends a poor prognosis.

In younger patients with good initial performance status, concomitant chemoradiotherapy should be used. However, with older patients and patients with poor initial performance, the interval between the end of chemotherapy and the beginning of radiotherapy should be individually defined; nevertheless. in the light of our results, it seems that the interval should not be longer than 19 days. Of course, factors that may determine the delay of radiotherapy after chemotherapy, such as the toxicity of chemotherapy, rapid tumor progression, patient refusal, or administrative reasons, should be taken into account. Similar problems with delay in therapy and the adverse effect of this delay on prognosis can be observed, for example, in patients qualified for postoperative adjuvant radiotherapy (not only in lung cancer, but also in head and neck cancers); a similar risk exists in the case of delay in the use of durvalumab after chemoradiotherapy for NSCLC stage III patients.

New and more available diagnostic abilities (PET, EBUS, EUS) allow us to recognize the appropriate stage of cancer disease, and the best treatment strategies may be used not only in relation to cancer advancement; matching the molecular type is also essential. This means the mainly personalized use of immune check point inhibitors, targeted therapies, etc., in modern oncology. However, the potential role of radical chemoradiotherapy, even sequential chemoradiotherapy, remains important for large groups of NSCLC patients.

The essential limits of this study should be kept in mind in the context of detailed interpretations of the results: i.e., the study's retrospective nature, the single-institution experience, the absence of correlation with the RECIST response to treatment, the absence of multivariate analysis including established prognostic factors (T stage, N stage, initial PS, etc.), and the lack of direct analysis of potential determinants of the interval between chemotherapy and radiotherapy. Moreover, due to the very large variety of chemoradiotherapy regimens used, no analysis of their impact on KPS was conducted. The next clinical problem is associated with the fact that KPS is a subjective scale and may be falsely elevated, especially before cancer therapy. However the decrease in KPS should be clearly noticed by oncologists. If the study's hypothesis is a correlation between worsening KPS and ineffective cancer therapy (insufficiently well-chosen), it is easy to conclude that the better the patient's baseline performance status, the greater the risk of performance-status deterioration during chemoradiotherapy (by 25% for every 10 on the Karnofsky scale).

## 5. Conclusions

The monitoring of performance status should be mandatory during chemoradiotherapy, as performance-status deterioration predicts a poor long-term prognosis. The risk of performance-status deterioration correlates with each day of the interval between chemotherapy and radiotherapy and is significantly higher in younger patients and in patients with better baseline performance status. Even if the results may be questionable due to the retrospective nature of the study, the main conclusion merits prospective further investigation.

Currently, it seems fully justified to monitor performance status because its deterioration can be considered for subsequent escalating or de-escalating treatment strategy and/or intensity during chemoradiotherapy in stage III of NSCLC.

**Author Contributions:** Conceptualization, M.O.-S., M.Z.-S., S.S., D.M.K.; methodology, M.O.-S., M.Z.-S., S.S., D.M.K.; software, M.O.-S., M.Z.-S., P.J., D.M.K.; validation M.K., D.M.K.; formal analysis, M.O.-S., M.Z.-S., S.S., D.M.K. investigation, M.O.-S., M.Z.-S., K.Z.; resources, M.O.-S., M.Z.-S.; data curation, M.O.-S., M.Z.-S.; writing—original draft preparation, M.O.-S., M.Z.-S., S.S.; writing—review and editing, All authors; visualization, M.O.-S., M.Z.-S., S.S., D.M.K.; supervision, M.K., D.M.K.; project administration, M.Z.-S., D.M.K.; funding acquisition, M.K., D.M.K.; All authors have read and agreed to the published version of the manuscript.

**Funding:** This research received no external funding.

**Institutional Review Board Statement:** The study was conducted according to the guidelines of the Declaration of Helsinki. The study was approved by the Bioethical Committee of the National Research Institute of Oncology in Poland (opinion number 70/2021).

**Informed Consent Statement:** The study was based on the retrospective collection and analysis of administrative data from the National Research Institute of Oncology in Poland.

**Data Availability Statement:** Data may be available upon reasonable request and with permission of the National Research Institute of Oncology in Warsaw (Poland).

**Conflicts of Interest:** Authors declare no conflict of interest directly associated with the study.

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
