# Peer review of "Performance-Status Deterioration during Sequential Chemo-Radiotherapy as a Predictive Factor in Locally Advanced Non-Small Cell Lung Cancer"

_curroncol, doi:10.3390/curroncol30020159_

Round 1

Reviewer 1 Report

The paper entitled Performance status deterioration during sequential chemo-2 radiotherapy as a prognostic factor in locally advanced non-3 small cell lung cancer has a very pragmatic objective by using an easy clinical tool, the Karnofsky performance status degradation after chemotherapy, as predictive biomarker for a retrospectively reviewed 196 patients stage III NSCLC cohort. Even that the results are highly questionable because of  biases that need clarification  (please see below) and  due to the retrospective nature, the conclusion merits prospective further investigation.

 Abstract: From a statistical point of view, please reconsider the terms PROGNOSTIC (independent of treatment, when no treatment or when different treatments are compared) related to OS and PREDICTIVE one related to PFS, in a single setting treatment scenario (sequential chemo-RT, for this paper). For me, although I understand implying a worse prognosis (just as a non-medical general speaking term), KPS decrease is (only) predictive for both worse PFS and OS for seq CT-RT

Introduction is too long and not directly related to the subject. Relevant info from physical exercise, cardiac function etc can be used in the discussion session. The first 3 paragraphs can be deleted without compromising the setting stage purpose of the introduction to the studied topic.

Even if cardiopulmonary function contributes at most to the KPS, other conditions are involved (significant weight loss or on contrary morbid obesity, significant neurologic impairment ….)

LIGHT BLUE- English rephrasing needed; RED-= avoid misspelling or redundancy; YELLOW-delete(?)

MEHODS:

-        state the nature of the study- for ex: retrospective, uni-institutional, observational study.

-        INCLUSION criteria for the selected patients are rather flu. Just by sending the reader to a previous study by the same group is not enough (Cancer, 2021, ref 12)

-        Time points of KPS assessment need clarification: where they only at the start and the end of tt? Or at any other points also – for ex at the end of chemo, at the start of RT ?

-        What was measured as KPS deterioration?  - For ex.  any decrease of KPS between previous defined points.

-        The sequential RT was without any radiosensitizing chemo (e.g.weekly platinum), right?

-        Toxicity scale used

RESULTS:

-        Fig 1 & 2 (OS & PFS) need better, self-explanatory presentation (numbers at risk according to timeline, mSV , HR, p …)

-        Tables: adding an explanation like “OR (odd ratio) below 1 relates to a better outcome” might be helpful

DISCUSSION:

KPS deterioration has potential biases- MULTIVARIATE Cox analysis (short RT < 28 days= palliative or stopped because of toxicity, progression?;  less chemo< 2 cycles seem to do better in terms of KPS deterioration, but is  unlikely to give a better outcome as to OS, PFS.

Addressing briefly that the nowadays new more available diagnostic abilities (PET, EBUS, EUS) and  treatment strategies in this group, i.e IO, EGRF, ALK targeted therapies etc still leave a great number of patients in this same and therefore still important scenario of seq CT-RT.

SHALL we strive to start RT asap, ideally in less than 19 days? The importance of the Interval until starting RT, which is the main conclusion of the paper warrants elaborate discussion and data dissection as this might be an important message. How often the RT was delayed because of chemo toxicity, progressive disease, patient refusal, administrative reasons etc? Other parallel examples of timeline of sequential treatments – e.g. with postop  adj RT for  locally advanced H&N operated patients, or with starting adj Durvalumab after CRT for NSCLC stage III patients can be exploited

Acknowledge the LIMITS of the study: retrospective, single institution…, NO PRO/s, No correlation with the RECIST response to treatment. No multivariate analysis including established prognostic factors (T stage, N stage, initial PS, etc), potential bias on the interval between CT and RT.etc.

CONCLUSION: - you might consider a “call to action” message at the end, such as: ….” …performance status, factors than can be considered for subsequent escalating or de-escalating treatment strategy and/or intensity”.

Author Response

The paper entitled Performance status deterioration during sequential chemo-radiotherapy as a prognostic factor in locally advanced non-small cell lung cancer has a very pragmatic objective by using an easy clinical tool, the Karnofsky performance status degradation after chemotherapy, as predictive biomarker for a retrospectively reviewed 196 patients stage III NSCLC cohort. Even that the results are highly questionable because of  biases that need clarification  (please see below) and  due to the retrospective nature, the conclusion merits prospective further investigation.

 Many thanks for the criticism. I have added the following text to conclusions

"Even if the results may be understood questionable due to the retrospective nature of the study, the main conclusion merits prospective further investigation".

 Abstract: From a statistical point of view, please reconsider the terms PROGNOSTIC (independent of treatment, when no treatment or when different treatments are compared) related to OS and PREDICTIVE one related to PFS, in a single setting treatment scenario (sequential chemo-RT, for this paper). For me, although I understand implying a worse prognosis (just as a non-medical general speaking term), KPS decrease is (only) predictive for both worse PFS and OS for seq CT-RT

 Many thanks for this suggestion. We agree and have corrected the abstract. Please see the sentence:

Deterioration of KPS during chemoradiotherapy was observed in 53 patients (27%) and had a negative predictive value for both worse progression free survival (HR=1.44; 95% CI: 1.03-1.99; p=0.03) and overall survival (HR=1.42; 95% CI: 1.02-1, 99; p=0.04).

Introduction is too long and not directly related to the subject. Relevant info from physical exercise, cardiac function etc can be used in the discussion session. The first 3 paragraphs can be deleted without compromising the setting stage purpose of the introduction to the studied topic.

We have moved the first 3 paragraphs to the discussion session.

Even if cardiopulmonary function contributes at most to the KPS, other conditions are involved (significant weight loss or on contrary morbid obesity, significant neurologic impairment ….)

We have added the following sentence to the introduction:

Worse KPS is determined by many factors like heart dysfunction, limitation in ventilation, decreased lung capacity, significant weight loss, anemia, morbid obesity or cachexia, neurological impairment etc.

MEHODS:

-        state the nature of the study- for ex: retrospective, uni-institutional, observational study.

We have defined the design of the study as:

The retrospective, uni-institutional, observational study as PhD study was planned for identification of all possible prognostic and predictive factors in locally advanced NSCLC treated by radical chemoradiotherapy.

-        INCLUSION criteria for the selected patients are rather flu. Just by sending the reader to a previous study by the same group is not enough (Cancer, 2021, ref 12)

We have added the inclusion criteria.

-        Time points of KPS assessment need clarification: where they only at the start and the end of tt? Or at any other points also – for ex at the end of chemo, at the start of RT ?

We have added the information:

Assessment of KPS was mandatory at least at the start of treatment before chemotherapy and at the end of radiotherapy.

-        What was measured as KPS deterioration?  - For ex.  any decrease of KPS between previous defined points.

We have added the information:

 KPS deterioration was understood as a decrease of at least 10 points between the first and last assessment.

-        The sequential RT was without any radiosensitizing chemo (e.g.weekly platinum), right?

Radiosensitizing chemotherapy had to be used before radiotherapy, there was a possible choice between: cisplatine-based regimen (PN: cisplatin + vinorelbine or PG: cisplatin + gemcitabine or PE cisplatin + etoposide), carboplatin-based regimen (carboplatin + vinorelbine or carboplatin + paclitaxel). 

-        Toxicity scale used

Toxicity was recognized in accordance with the Common Terminology Criteria for Adverse Events (CTCAE)

RESULTS:

-        Fig 1 & 2 (OS & PFS) need better, self-explanatory presentation (numbers at risk according to timeline, mSV , HR, p …)

The quality of figures was improved.

-        Tables: adding an explanation like “OR (odd ratio) below 1 relates to a better outcome” might be helpful

We have added the following explanation for OR as legend of table:

LEGEND: OR - odds ratio, here determines the ratio of the chance of KPS deterioration, a value above 1 determines how much greater (a value below 1 determines how much lower) the chance of KPS deterioration occurring in patients with a particular prognostic factor.

DISCUSSION:

KPS deterioration has potential biases- MULTIVARIATE Cox analysis (short RT < 28 days= palliative or stopped because of toxicity, progression?;  less chemo< 2 cycles seem to do better in terms of KPS deterioration, but is  unlikely to give a better outcome as to OS, PFS.

Our observation is a hypothetical study. The hypothesis assumes that the deterioration of KPS is due to the lower effectiveness of cancer treatment. This is confirmed by the fact that the deterioration of KPS correlates with each day of treatment extension, and precisely with each day between the end of chemotherapy and the start of radiotherapy. Our data clearly show that the younger the patient and the better the initial performance status, the greater the chance of worsening KPS because sequential chemoradiotherapy with a long time between chemotherapy and radiotherapy is not a sufficiently intensive cancer treatment for such patients.

Addressing briefly that the nowadays new more available diagnostic abilities (PET, EBUS, EUS) and  treatment strategies in this group, i.e IO, EGRF, ALK targeted therapies etc still leave a great number of patients in this same and therefore still important scenario of seq CT-RT.

The nowadays new more available diagnostic abilities (PET, EBUS, EUS) allow us to recognize appropriate stage of cancer disease and the best treatment strategies may be used in relation not only to cancer advancement but matching the molecular type is also essential. It means mainly personalized use of immune check points inhibitors, targeted therapies etc. in modern oncology. However the potential role of radical chemoradiotherapy even sequential stay still important scenario for large group of NSCLC patients.

SHALL we strive to start RT asap, ideally in less than 19 days? The importance of the Interval until starting RT, which is the main conclusion of the paper warrants elaborate discussion and data dissection as this might be an important message. How often the RT was delayed because of chemo toxicity, progressive disease, patient refusal, administrative reasons etc? Other parallel examples of timeline of sequential treatments – e.g. with postop  adj RT for  locally advanced H&N operated patients, or with starting adj Durvalumab after CRT for NSCLC stage III patients can be exploited

In younger patients with good initial performance status, concomitant chemoradiotherapy should be used. However, the older the patient and initially in poorer performance, the interval between the end of chemotherapy and the beginning of radiotherapy should be individually defined, but it seems in the light of our results that it should not be longer than 19 days. Of course, factors that may determine the delay of radiotherapy after chemotherapy, such as the toxicity of chemotherapy, rapid tumor progression, or even patient refusal, administrative reasons, should be taken into account. Similar problems with delay in therapy and the adverse effect of this delay on prognosis can be observed, for example, in patients qualified for postoperative adjuvant radiotherapy (not only in lung cancer, but also in head and neck cancers), a similar risk exists in the case of delay in the use of durvalumab after chemoradiotherapy for NSCLC stage III patients.

Acknowledge the LIMITS of the study: retrospective, single institution…, NO PRO/s, No correlation with the RECIST response to treatment. No multivariate analysis including established prognostic factors (T stage, N stage, initial PS, etc), potential bias on the interval between CT and RT.etc.

We have added the limits of the study.

CONCLUSION: - you might consider a “call to action” message at the end, such as: ….” …performance status, factors than can be considered for subsequent escalating or de-escalating treatment strategy and/or intensity”.

We have added the following "call to action":

However, today it seems fully justified to monitor of performance status because deterioration of KPS can be considered for subsequent escalating or de-escalating treatment strategy and/or intensity during chemoradiotherapy in stage III of NSCLC.

Reviewer 2 Report

I have read and reviewed the revised manuscript entitled “Performance statrus deterioration during sequential chemoradiotherapy as a prognostic factor in locally advanced non-small cell lung cancer”. The concept is quite good and intriguing. The results are even more interesting. However, the methods section is underwritten and information about chemotherapy regiments and the 196 patients is lacking. I hope the points below will help the authors improve the manuscript.

Major Points:

1.     A Table with the demographic breakdown of the group would be of benefit. This should show number of male/female, comorbid conditions, age (average and range), smoking status, pack years, number and percentage of the complications, etc… hinting at this data in Table 3 requires more information.

2.      As this manuscript makes a number of points about the value of sequential therapy over concurrent, the authors do not reference RTOG 9410. While this paper included surgery as part of the treatment, the effects of induction therapy are well described including the lower rate of pulmonary toxicity in the concurrent groups compared to the sequential. A more robust discussion

3.     The authors do not provide any information about the chemoradiotherapy regiments. What was the dose of radiation? What were the common chemotherapy regiments? Did patients who received certain regiments have less reduction in KPS?

4.     The data gets a bit tricky. Do the authors conclude that a patient with a baseline KPS of 100 that drops to 90 has a 7-fold increased risk of decreased OS and PFS compared to a patient that has a baseline KPS of 90 that does not change? Or more importantly, a patient with a baseline of 80 that does not change? Is it possible that KPS is too subjective, and the baseline values are falsely elevated? Did the authors consider using ECOG scores?

5.     Why was there a delay in starting radiotherapy? Could it be that these patients did not tolerate the chemotherapy or had a significant reduction in KPS prior to radiation which caused the delay? Then a delay in radiation portends a poor prognosis regardless of if they got the radiation.

6.     In the DISCUSSION (starting line 174) the authors that young patients with excellent KPS should not get sequential chemoradiotherapy as it results in ineffective cancer therapy and thus unfavorable OS and PFS. Do we know the cumulative dose of chemotherapy and if the patients completed radiation therapy? These would be markers of “inadequate cancer therapy”. If the drop in KPS caused dose reduction or fewer cycles of chemo, then the authors should present that data.

7.     An explanation of how and why the authors chose 19 days and 42 days as their timepoints for the variables of time from end of chemotherapy to the start of radiation. What was the result of those between 20 and 41 days?

Minor Points:

1.     Figure 1 should have in the legend identification of both the red and blue lines and not just one. Also, the notes about “complete” and “censored” can also go in the figure legend below the graph and not in the space for the graph title.

2.     Also in Figure 1, the graph title could be improved. Consider make Figure 1A: Overall Survival and for Figure 1B: Progression Free Survival. The reason is that the y-axis is not OS or PFS, but percent alive.

3.     I recognize that in Europe a comma is used to separate the ones column and tenths column. However, in English language and nearly all medical journals a “period” should be utilized. (e.g. replace 0,82 with 0.82 – p-value for female in Table 1). This holds for the entire paper. The authors already did this in the written section of the results, they need to keep the same formatting in that Tables.

4.     Please check the journals rules of formatting the references. I am not sure what it is but cannot imagine that citation 24 needs all 30+ authors listed. Then apply the rules to the entire reference section.

Author Response

I have read and reviewed the revised manuscript entitled “Performance statrus deterioration during sequential chemoradiotherapy as a prognostic factor in locally advanced non-small cell lung cancer”. The concept is quite good and intriguing. The results are even more interesting. However, the methods section is underwritten and information about chemotherapy regiments and the 196 patients is lacking. I hope the points below will help the authors improve the manuscript.

We would like to thank Reviewer for all help in improving quality of our manuscript.

Major Points:

  1. A Table with the demographic breakdown of the group would be of benefit. This should show number of male/female, comorbid conditions, age (average and range), smoking status, pack years, number and percentage of the complications, etc… hinting at this data in Table 3 requires more information.

We have added Table 1. Table 1. Characteristics of patients included to the study.

We have added the explanation For Table 3 (new Table 4).

  1. As this manuscript makes a number of points about the value of sequential therapy over concurrent, the authors do not reference RTOG 9410. While this paper included surgery as part of the treatment, the effects of induction therapy are well described including the lower rate of pulmonary toxicity in the concurrent groups compared to the sequential. A more robust discussion

We have extended the discussion by adding the text about RTOG 9410.

  1. The authors do not provide any information about the chemoradiotherapy regiments. What was the dose of radiation? What were the common chemotherapy regiments? Did patients who received certain regiments have less reduction in KPS?

We have added the information about chemotherapy regimens and the dose of radiation.

Due to the very large variety of chemoradiotherapy regimens used, no analysis of their impact on KPS was conducted. We added this information to the LIMITS of the study.

  1. The data gets a bit tricky. Do the authors conclude that a patient with a baseline KPS of 100 that drops to 90 has a 7-fold increased risk of decreased OS and PFS compared to a patient that has a baseline KPS of 90 that does not change? Or more importantly, a patient with a baseline of 80 that does not change? Is it possible that KPS is too subjective, and the baseline values are falsely elevated? Did the authors consider using ECOG scores?

We agree that KPS and ECOG are subjective scales and may be falsely elevated especially before cancer therapy. However we believe that decrease in KPS can be clearly noticed by oncologists. If the study's hypothesis is correlation between worsening KPS and ineffective cancer control, it is easy to adopt that  the better the patient's baseline performance status, the greater the risk of its deterioration during chemoradiotherapy (by 25% for every 10 on the Karnofsky scale).

We have added the following text to the LIMITS of the study:

The next clinical problem is associated with the fact that KPS is the subjective scale and may be falsely elevated especially before cancer therapy. However the decrease in KPS should be clearly noticed by oncologists. If the study's hypothesis is a correlation between worsening KPS and ineffective cancer therapy (insufficiently well-chosen), it is easy to adopt that  the better the patient's baseline performance status, the greater the risk of its deterioration during chemoradiotherapy (by 25% for every 10 on the Karnofsky scale).

  1. Why was there a delay in starting radiotherapy? Could it be that these patients did not tolerate the chemotherapy or had a significant reduction in KPS prior to radiation which caused the delay? Then a delay in radiation portends a poor prognosis regardless of if they got the radiation.

Yes exactly. The oncologists decided about a delay in starting radiotherapy. Among reasons the most important were: the patients did not tolerate the chemotherapy or had a significant reduction in KPS already before radiation. Such situation can expain each day of a delay in radiation portends a poor prognosis.

  1. In the DISCUSSION (starting line 174) the authors that young patients with excellent KPS should not get sequential chemoradiotherapy as it results in ineffective cancer therapy and thus unfavorable OS and PFS. Do we know the cumulative dose of chemotherapy and if the patients completed radiation therapy? These would be markers of “inadequate cancer therapy”. If the drop in KPS caused dose reduction or fewer cycles of chemo, then the authors should present that data.

The drop in KPS did not depend on duration of chemotherapy, number of cycles of chemotherapy, type of chemotherapy regimen, duration of radiotherapy (Table 3 = new Table 4). Inadequate cancer therapy meant too long delay between the end of chemotherapy and start of radiotherapy.

  1. An explanation of how and why the authors chose 19 days and 42 days as their timepoints for the variables of time from end of chemotherapy to the start of radiation. What was the result of those between 20 and 41 days?

We have added the information: The values of the lower and upper quartiles were used as prognostic factors.

Minor Points:

  1. Figure 1 should have in the legend identification of both the red and blue lines and not just one. Also, the notes about “complete” and “censored” can also go in the figure legend below the graph and not in the space for the graph title.

Figure was generated by Statistica software. The place of legend for “complete” and “censored” was prepared automatic by this program.

We corrected the explanation for the lines.

  1. Also in Figure 1, the graph title could be improved. Consider make Figure 1A: Overall Survival and for Figure 1B: Progression Free Survival. The reason is that the y-axis is not OS or PFS, but percent alive.

We deleted OS, PFS on the figures.

I would prefer to present Figure 1 and 2 because our primary observation point is OS and secondary PFS.

  1. I recognize that in Europe a comma is used to separate the ones column and tenths column. However, in English language and nearly all medical journals a “period” should be utilized. (e.g. replace 0,82 with 0.82 – p-value for female in Table 1). This holds for the entire paper. The authors already did this in the written section of the results, they need to keep the same formatting in that Tables.

We corrected the Tables according to your comment.

  1. Please check the journals rules of formatting the references. I am not sure what it is but cannot imagine that citation 24 needs all 30+ authors listed. Then apply the rules to the entire reference section.

The section of references will be corrected.

Reviewer 3 Report

This retrospective study looks at the very important impact of performance status on outcome of curatively intended sequential chemo radiation for locally advanced NSCLC

The intro is confusing as it talks about exercise and cardiofunction  which is not strictly the topic of study. Also cardiotoxicity is only capturing a bit of the KPS problem. So intro will have to be rewritten

The endpoint is survival which is relevant. The study is of adequate size.

Decline in KPS affects outcome as expected, what was not expected is that that baseline KPS was not related to outcome. Further it is surprising that good KPS patients have the biggest drop in KPS, I would expect that those with less than 90 to be more vulnerable and thus subject to more drop. Maybe this should be discussed.

Not sure I understand the relationship with age and drop in KPS, authors need to explain better.

Discussion is good and exhaustive and not with the cardiology focus of the introduction.

The authors suggest that findings can be used for a personal algorithm for concurrent and seq chemo radiotherapy, but with only data for concurrent it is hard to understand the value of this and the authors should consider to revise to limit suggestion to sequential treatment.

Author Response

This retrospective study looks at the very important impact of performance status on outcome of curatively intended sequential chemo radiation for locally advanced NSCLC

We would like to thank the Reviewer for the valuable opinion on our study.

The intro is confusing as it talks about exercise and cardiofunction  which is not strictly the topic of study. Also cardiotoxicity is only capturing a bit of the KPS problem. So intro will have to be rewritten

The section of INTRODUCTION has been rewritten.

The endpoint is survival which is relevant. The study is of adequate size.

Thanks you for your positive opinion.

Decline in KPS affects outcome as expected, what was not expected is that that baseline KPS was not related to outcome. Further it is surprising that good KPS patients have the biggest drop in KPS, I would expect that those with less than 90 to be more vulnerable and thus subject to more drop. Maybe this should be discussed.

We've added some thoughts to the discussion of how we interpret our results. We describe sequential chemoradiotherapy that is suitable for the elderly and those with disabilities (personalized therapy for them). On the other hand, sequential therapy is too ineffective for younger people and in a good initial performance status. In our study, a decrease in KPS is a marker of ineffective therapy (predictive factor). Please read our changes highlighted in blue font.

Not sure I understand the relationship with age and drop in KPS, authors need to explain better.

We added our explanation and interpretation of the results. Please our discussion.

Discussion is good and exhaustive and not with the cardiology focus of the introduction.

Thank you for the positive opinion.

The authors suggest that findings can be used for a personal algorithm for concurrent and seq chemo radiotherapy, but with only data for concurrent it is hard to understand the value of this and the authors should consider to revise to limit suggestion to sequential treatment.

We agree. Please see the new text at the end of our discussion.

Round 2

Reviewer 1 Report

Congrats for the improvement.

W/o being mandatory, native English editing, ideally with medical background, will ease the lecture.

Reviewer 2 Report

I have read and reviewed the revised manuscript entitled “Performance status deterioration during sequential chemoradiotherapy as a predictive factor in locally advanced non-small cell lung cancer”. I would like to thank the authors for their revisions. It is significantly improved. A few points to consider to help with readability.

Points:

1.     In the revised first paragraph, ending a sentence with “etc” should not be done. My recommendation would be to alter the sentence and end it with “… neurological impairment, and older age.”

2.     “uni-institutional” could be single institution.

3.     No need to say that this was a PhD thesis/project.

4.     My apologies, it appears that the journal does want all the authors listed per MDPI Chicago Style. Thank you for looking into it.